# IL-18 production is required for the generation of a Th1 response during experimental chromoblastomycosis

Lucas Golçalves Ferreira, Sandro Rogério de Almeida[ID]*¤

School of Pharmaceutical Sciences, Department of Clinical e Toxicological Analysis, University of São Paulo, São Paulo, Brazil

¤ Present Address: Faculdade de Ciências Farmacêuticas, Departamento de Análises Clínicas e Toxicológicas, Universidade de São Paulo, Avenida Prof. Lineu Prestes, 580, Bloco 17, São Paulo, Brazil.
* sandroal@usp.br

## Abstract

Chromoblastomycosis is a chronic fungal infection characterized by the formation of granulomatous lesions in the skin and subcutaneous tissues that begins after inoculation trauma. The disease is more frequently observed in tropical countries such as Brazil. Important studies have been shown a predominantly cell-mediated immune response during chromoblastomycosis. Results from our laboratory showed that Th1 responses are essential to induce protection during chromoblastomycosis. IL-18 is primarily produced by macrophages and is known to induce the production of IFNγ, a cytokine associated with Th1 cell activation. Once produced, IL-18 acts to promote Th1 cell differentiation and activation. Th1 cells, in turn, secrete cytokines such as IFNγ, which are critical for the elimination of intracellular pathogens, including fungi. IFNγ enhances the fungicidal activity of macrophages, promotes the development of antifungal effector mechanisms, and contributes to the containment of fungal growth. Our results indicate that *F. pedrosoi* is sensed by the NLRP3 inflammasome, which induces caspase-1 activation and production of IL-18. Moreover, IL-18 plays a crucial role in activating Th1 cells and controlling fungal loads during chromoblastomycosis. Further research into the mechanisms underlying IL-18-mediated immunity may lead to the development of novel therapeutic approaches for the treatment of this chronic fungal infection.

## Introduction

Chromoblastomycosis is a chronic fungal infection primarily affecting the skin and subcutaneous tissues. It is caused by several species of dematiaceous fungi such as *Cladophialophora*, *Exophiala*, *Fonsecaea*, *Phialophora*, and *Rhinocladiella*. *F. pedrosoi*, *F. monophora*, and *F. nubica* represent three cryptic entities that could induce the same disease [1–4]. *Fonsecaea pedrosoi* and *F. nubica* are associated with muriform

**Data availability statement:** All relevant data are within the paper and its Supporting Information files.

**Funding:** This work was supported by São Paulo Research Foundation (FAPESP): Grant FAPESP 2016/04729-3. The funders had no role in study design, data collection and analysis, decision to publish, or preparation of the manuscript.

**Competing interests:** The authors have declared that no competing interests exist.

cell formation in chromoblastomycosis, whereas *F. monophora* could be involved in dissemination to the brain and other organs with hyphae in tissue [5]. Recently, some cases have been confirmed caused by *Fonsecaea monophora* [6,7]. These fungi are found in soil, decaying vegetation, and wood, particularly in tropical and subtropical regions.

The infection typically begins following traumatic inoculation of fungal spores into the skin, often through minor injuries like cuts, scratches, or thorn pricks. Once inoculated, the fungi establish a chronic infection characterized by the formation of verrucous or nodular lesions on exposed areas of the body, most commonly the lower extremities. [8,9]. The severity of disease is graded according to number, extension, and dissemination of the lesions as described by Queiro-Telles *et al*. [10].

In general, the treatment of chromoblastomycosis is a therapeutic challenge because various methods and antimicrobials are used in attempts to cure or control the disease, but most of them fail. There are many cases of relapse and few reports of ultimate cures [4,11].

The immune response to chromoblastomycosis involves both innate and adaptive components. Our laboratory showed that PBMC from patients with severe forms of chromoblastomycosis failed to proliferate *in vitro* after induction with chromoAg and produced high levels of IL-10 and low levels of IFN-γ [12]. In contrast, cells from patients with mild forms of the disease efficiently induced T-cell proliferation and IFN-γ production. These results suggest that the Th1 response is essential to induce protective immunity during chromoblastomycosis*.*

The recognition of pathogens is a severe challenge to the innate immune system due to the enormous variability and characteristically high rate of mutation of microorganisms. However, the innate immune system has evolved receptors, which target molecular structures common to large groups of pathogens or produced by pathogens, not the host. These structures are called pathogen-associated molecular patterns (PAMPs). The PAMPs are recognized by receptors called pattern recognition receptors (PRRs). Members of several family proteins function as PRRs and are expressed in cells responsible for the first line of defense of the body, such as macrophages and dendritic cells [13]. Caspase-1 is produced as a zymogen, called procaspase-1, cleaved into two subunits of 20-kDa (p20) and 10 kDa (p10). This event transforms the zymogen into an active enzyme that can cleave proteins at specific regions, such as the IL-1β and IL-18 cytokines [14,15]. Interleukin-18 (IL-18) is a pro-inflammatory cytokine that plays a crucial role in regulating immune responses, particularly those involving T cells and natural killer (NK) cells. It is a member of the interleukin-1 (IL-1) superfamily and is produced by various cell types, including macrophages, dendritic cells, epithelial cells, and endothelial cells, in response to microbial products, inflammatory stimuli, or other cytokines.

IL-18 is synthesized as an inactive precursor protein (pro-IL-18) and requires proteolytic processing by caspase-1, which is activated by multiprotein complexes known as inflammasomes, to become biologically active [16]. The primary function of IL-18 is to induce the production of interferon-gamma (IFNγ), another important pro-inflammatory cytokine, mainly from T cells and NK cells. This makes IL-18 a

potent inducer of Th1 responses, which are characterized by the activation of CD4+ T cells to produce IFNγ and promote cell-mediated immunity against intracellular pathogens, including bacteria, viruses, and fungi.

Our laboratory [17] showed that macrophages infected by *F. pedrosoi* and *R. aquaspersa*, the etiological agents of chromoblastomycosis, induced the production of inflammatory cytokines such as IL-1β and TNF-α. De Castro et al. 2017, showed that *F. pedrosoi* hyphae, induce IL-1β secretion in bone marrow-derived dendritic cells and macrophages. IL-1β production was NLRP3-dependent inflammasome activation, which required potassium efflux, reactive oxygen species production, phagolysosomal acidification, and cathepsin B release as triggers. However, there is no data on the importance of IL-18 in activating Th cells during chromoblastomycosis. Our results indicate that *F. pedrosoi* is sensed by the NLRP3 inflammasome, which induces caspase-1 activation and a proinflammatory response. Moreover, we showed that IL-18 is required to activate the Th1 response and control fungal loads during chromoblastomycosis.

## Materials & methods

### Microorganism

The isolate of *F. pedrosoi* (CBS125763) was used for the present investigation. Stock cultures were maintained on Sabouraud-dextrose-agar (SDA), under oil at 4$^\circ$C, with transfer every six months.

### Conidia of *F. pedrosoi*

To obtain large numbers of conidia, fungal fragments from the Sabouraud medium were scraped off the agar and incubated in Potato-broth medium, which consisted of 50 g of potatoes, 5 g of glucose (MERCK – Germany), and 500 mL of distilled water at 25°C under constant rotation for four days. The fungal suspension was then filtered through a sterile Whatman #1 filter to remove hyphal fragments but not microconidia. The conidia were washed with PBS and counted in a hemocytometer.

### Animals

Wild-type male C57BL/6 mice, 8–12 weeks old, were provided from the Faculty of Pharmaceutical Sciences, University of São Paulo (USP) animal facilities. Male C57BL/6 mice, 8–12 weeks old, knocked out individually for NLRP3, IFN-γ, Caspase-1/-11, or IL-18 were obtained from the animal facilities at the School of Medicine of Ribeirão Preto (University of São Paulo). The Institutional Ethics Committee previously approved the experimental protocols involving animals for Animal Care and Research at the Faculty of Pharmaceutical Sciences (Protocols n$^\circ$: 057). The *in vivo* experiments were carried out following the recommendations of the ARRIVE Guidelines and the Guide for the Care and Use of Laboratory Animals of the National Institutes of Health. The animals were maintained in an SPF environment and housed in temperature-controlled rooms at 23–25 °C with food and water ad-libitum throughout the experiments. The euthanasia procedure was performed according to the American Veterinary Medical Association Guidelines for the Euthanasia of Animals (2020), using the overdose of ketamine and xylazine method.

### Bone Marrow-Derived Macrophages (BMDMs)

BMDMs were obtained as described [18]. Briefly, bone marrow from the femurs of C57BL/6 mice was collected and cultured in $R_{20/30}$ media, which is RPMI-1640 medium supplemented with 20% FBS, 30% L929-cell conditioned medium (LCCM) and gentamicin (40 mg/L), at 37°C in a 5% $CO_2$ atmosphere. After four days, the cells were supplemented with the same medium and incubated for three more days. Macrophages were collected and incubated in $R_{10/5}$ media, RPMI-1640 medium supplemented with 10% FBS, 5% LCCM and gentamicin (40 mg/L), at 37°C in a 5% $CO_2$ atmosphere. To validate the participation of K+ efflux or ROS production on the activation of the inflammasome, the BMDMs were treated for 2 h before the infection with 130 mM potassium chloride or 5 mM *N*-acetylcysteine (NAC) or 100 μM APDC (2R,4R)-4

aminopyrrolidine-2,4-dicarboxylic acid. To evaluate the importance of endosomal acidification on the activation of inflammasomes, the BMDMs were treated for 2h before the infection with 50 µM of cathepsin B inhibitor CA-074Me or 250nM of inhibitor bafilomycin.

### Interaction between BMDMs and *F. pedrosoi in vitro*

BMDMs were plated over glass slides in 24-wells plates and kept overnight at 37°C and 5% $CO_2$ to allow cell adhesion. *F. pedrosoi* conidia produced as described above were diluted in fresh $R_{10/5}$ media and incubated with the BMDMs at a MOI of 1:5. The systems were maintained at 37°C and 5% $CO_2$ for 12 hours. After 12h, the supernatants were collected and centrifuged at 3000rpm for 10 minutes and stored for cytokine measurements.

### Infection assays

In this work we used the murine chronic model of chromoblastomycosis described by Cardona-Castro et al. 1999 [19]. The C57BL/6 mice, received an intraperitoneal (i.p.) injection of 1ml of PBS (the control) or $2 \times 10^7$ conidia of *F. pedrosoi*. After 14 days post-infection, the animals were sacrificed, and the spleen and liver were removed for fungal quantification and measurement of cytokines in the organs' supernatants.

### Colony Forming Units (CFUs)

To evaluate the *in vivo* survival of fungal infections, the spleen and liver were macerated in PBS surface seeded onto plates containing Sabouraud agar. It incubated at 37 ° C for 14 days to count the colony-forming units (CFUs).

### Quantification of cytokines

All cytokines were measured by ELISA using monoclonal antibody capture and detection against cytokine concentrations recommended by the manufacturer (R & D Systems).

### Flow cytometry

The cells from the spleen and liver were stained with Abs specific for the surface molecules CD3 and CD4 and for the intracellular cytokine IFN-γ. For the intracellular cytokine staining, the cells were previously permeabilized using 1× PBS containing 1% FBS, 0.1% sodium azide, and 0.2% saponin. The cellular data were acquired using a FACSCanto II flow cytometer, and data were analyzed using FlowJo software (Tree Star, Ashland, OR).

### Statistical analysis

Data were expressed as mean+s.e.m. and analyzed in the software GraphPad Prism (version 5.00 for Windows, GraphPad Software, San Diego California USA, www.graphpad.com). For data analysis, the following statistical tests were used: Two-way ANOVA and Bonferroni post test, and Paired t-test. Differences were considered statistically significant at $p<0.05$.

## Results

### IL-18 production by macrophages in the presence of *F. pedrosoi* is dependent on NLRP3 and Caspase-1

First, we evaluated the production of anti and proinflammatory cytokines by BMDM in the presence of F. pedrosoi. The results showed that *F. pedrosoi* induces the production of TNF-α, IL-1-β and IL-18 by macrophages from C57BL/6 mice. No significant differences were observed in the secretion of IL-10 and IL-12 (Fig 1A). The next step was to evaluate the participation of caspase-1 and NLRP3 in IL-18 secretion by macrophages in the presence of *F. pedrosoi*. For this, the macrophages from wild-type C57BL/6 mice and caspase-1 and NLRP3 knockouts C57BL/6 mice were infected for 12

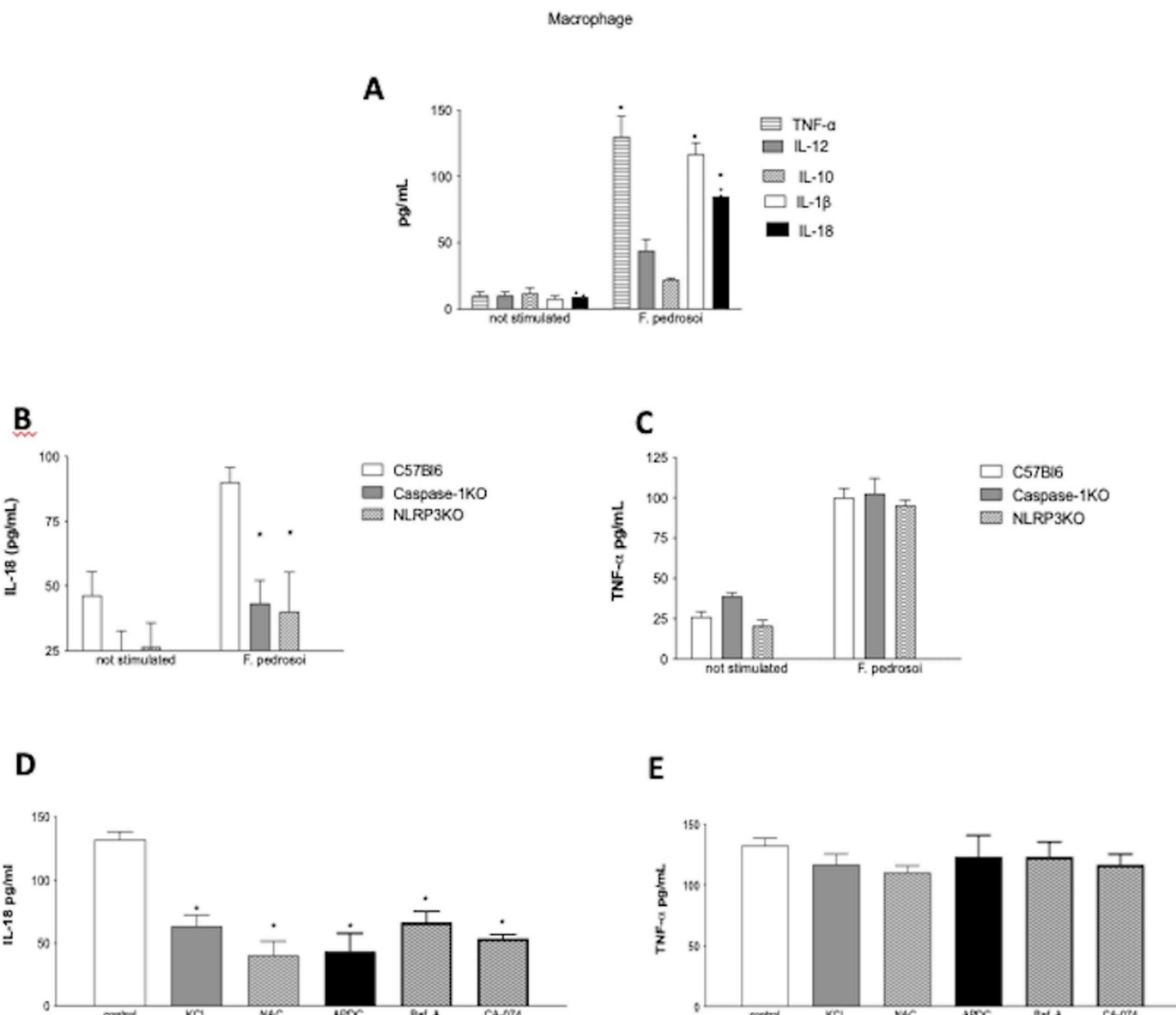

**Fig 1. NLRP3 and Caspase-1 deficiency reduces IL-18 secretion by BMDMs, but not TNF- α.** BMDMs from C57BL/6 mice were incubated with *F. pedrosoi* conidia for 12 h, and MOI 1:5, and TNF-**α**, IL-1β, IL-12, IL-10, and IL-18 were detected by ELISA (A). BMDMs from NLRP3-/- or Caspase-1/11-/- C57BL/6 mice were incubated with F. pedrosoi conidia for 12 h and MOI 1:5. (B) IL-18 production by BMDMs from NLRP3-/- or Caspase-1/11-/- C57BL/6 mice (C) TNF-α. NLRP3 inflammasome activation by F. pedrosoi requires lysosomal acidification, K+ efflux, and the release of cathepsin B. BMDMs from C57BL/6 mice were incubated with KCL 130 mM (block of K+ efflux), NAC 5 mM or APDC 100μM (ROS depletion), or Bafilomycin A 250 nM or CA-074 50 μM (lysosomal acidification), or without inhibitor (control) for 2 hours before the infection with *F. pedrosoi* and IL-18 (D) and TNF-**α** (E) determined. Results are expressed as mean ± SEM. Two-way ANOVA and Bonferroni post test: * p<0.05 when compared with control. Cytokine data pooled from three independent experiments performed in triplicate wells each are shown.

hours, and IL-18 production was monitored. The activation of caspase-1 in response to *F. pedrosoi* was dependent on NLRP3 because the secretion of IL-18 from macrophages of NLRP3 and caspase-1 KO animals was significantly lower when compared with that of wild-type macrophages (Fig 1B). As expected, TNF-α production by macrophages infected by *F. pedrosoi* is independent of caspase-1 and NLRP3, because no difference in TNF-α production was observed between WT and KO macrophages (Fig 1C). Our results showed that $K^+$ efflux, lysosomal acidification, cathepsin B activity, and ROS production interfere with IL-18 secretion, but not with TNF-α production (Fig 1D and 1E).

### The control of chromoblastomycosis is dependent on IFN-γ

Our laboratory showed that patients with a severe form of chromoblastomycosis presented a low level of IFN-γ, but on the other hand, patients with a mild form presented a high level of IFN-γ [12]. This result suggests that IFN-γ is essential to control the disease. To confirm the importance of IFN-γ in the control of infection, we conducted experiments using C57BL/6 mice KO to IFN-γ. As expected, our results showed that animals deficient in IFNγ production had a high fungal burden when compared with control animals (Fig 2).

### IL-18 is essential for the control of chromoblastomycosis

The next step was to evaluate the participation of IL-18 in the control of infection by *F. pedrosoi*. Then, we analyzed *F. pedrosoi* infections in C57BL/6 mice KO mice lacking functional *IL-18* genes. Our results showed that IL-18 is essential for the control of infection because the absence of this cytokine results in a severe infection (Fig 3A). When we analyzed the cytokine production, it was possible to observe a significant decrease in IFN-γ output in the organs of KO mice when compared with normal mice (Fig 3B).

### Animals deficient in IL-18 presented decreased Th1 cells

Several papers have shown that IL-12 and IL-18 are essential for inducing a vigorous Th-1 response [20–23]. In order to evaluate the importance of IL-18 in inducing Th1 cells, we analyzed the percentage of IFNγ–CD4+ T cells in the organs of infected animals and deficient to IL-18. Our results show a significant decrease of Th1 cells in the organs of C57BL/6 KO mice suggests that IL-18 is implicated in the induction of a Th1 response and control of chromoblastomycosis (Fig 4).

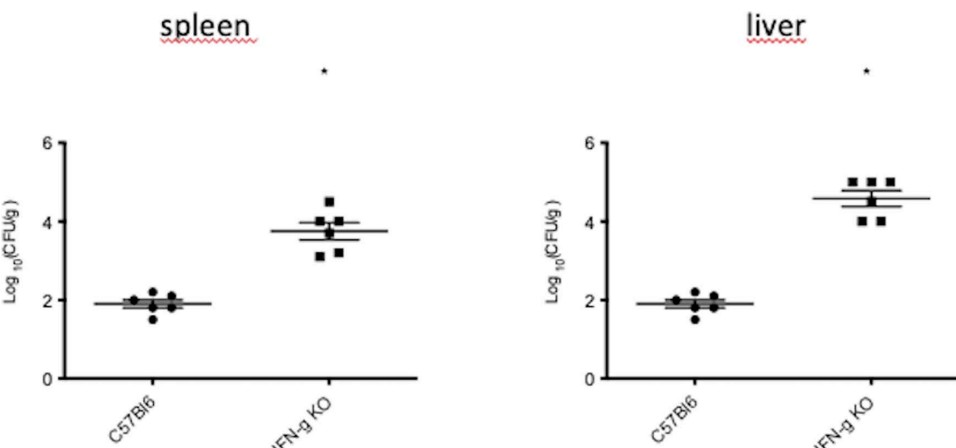

**Fig 2. Systemic chromoblastomycosis in mice deficient to IFN- γ.** Animals IFN-γ KO were infected i.p. with 2 x10⁷ conidia of *F. pedrosoi,* and after 14 days post-infection, the spleen and liver were removed, and fungal burden was determined by CFU. Data expressed as mean ± SEM. CFU data were pooled from three independent experiments performed. * $p<0.05$ when compared with WT mice (C57Bl6).

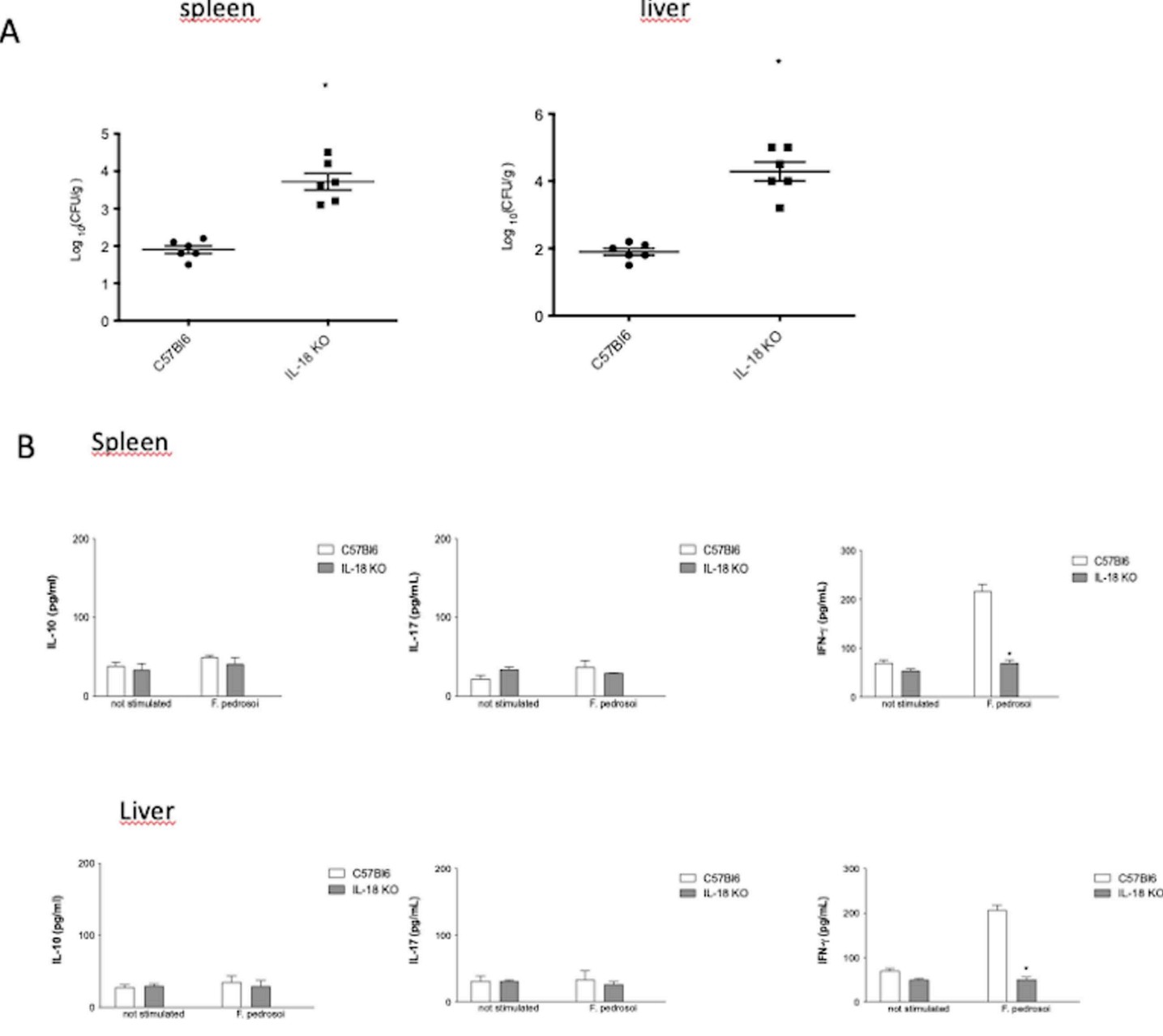

**Fig 3. A-Chromoblastomycosis in mice deficient in IL-18.** Animals were infected i.p. with $2 \times 10^7$ conidia of *F. pedrosoi,* and after 14 days post-infection, the spleen and liver were removed, and fungal burden was determined by CFU. Data expressed as mean ± SEM. CFU data were pooled from three independent experiments performed. * p<0.05 when compared with WT mice (C57Bl6). Fig 3 B- **Mice deficient in IL-18 show decreased production of IFN-γ in response to *F. pedrosoi*.** Animals were infected i.p. with *F. pedrosoi* conidia, and cytokines levels (IFN-γ; IL-10 and IL-17) were determined in spleen and liver homogenates 14 days post-infection.

## Discussion

The expansion of a subpopulation of T-cells (Th1, Th2, and Th17) for infection control is significant, as specific pathogens are more effectively controlled by Th1 and Th17 cells and others more effectively by a Th2 response. Recent studies have shown that cellular Th1 and Th17 responses induced by *Candida albicans* prevent the spread of disease [24–27].

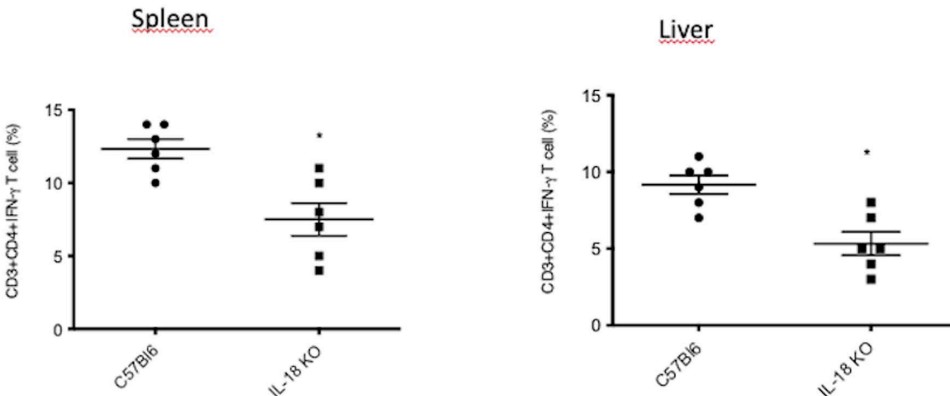

**Fig 4. Mice deficient in IL-18 show decreased IFN-γ-CD4 T cells in response to _F. pedrosoi_.** After infection, the cells from the spleen and liver were stained with Abs specific for the surface molecules CD3 and CD4 and the intracellular cytokine IFN-γ. The cellular data were acquired using a FACS-Canto II flow cytometer, and data were analyzed using FlowJo software. Two-way ANOVA and Bonferroni posttest: * p<0.05 compared to WT mice.

In our study, we found that _F. pedrosoi_ triggered the activation of caspase-1 and the production of IL-18 in vitro. Additionally, we observed that IL-18 secretion was reduced in the absence of NLRP3. NLRP3 has been associated with the response against other fungal pathogens, such as _Candida albicans_ [28], _Aspergillus fumigatus_ [29], _Paracoccidioides brasiliensis_ [30] and dermatophytes such as _Trichophyton rubrum_ [18] and _Microsporum canis_ [31]. In accordance with recent publications, our results show that _F. pedrosoi_ also induces IL-18 through this inflammasome and support the notion that NLRP3 may be a critical fungal sensor. Different processes associated with intracellular perturbations have been implicated in NLRP3 inflammasome activation, including K⁺ cation efflux, ROS generation, lysosomal acidification, and the release of cathepsin B into the cytosol. When the macrophages were treated with NAC or APDC a significant decrease in cytokine production was observed, demonstrating that neither K⁺ efflux blockade nor ROS depletion interferes with IL-18 secretion. In the same way, when BMDMs were treated with bafilomycin, which inhibits the vacuolar H⁺ ATPase, or CA-074, which inhibits cathepsin B activity, was observed a significant decrease of IL-18 secretion.

It is essential to point out here that the systemic mouse model of chromoblastomycosis used in this work is not an accurate representation of the human subcutaneous infection, but it is thought to be the best model for studying the chronic nature of this disease [19]. Moreover, the systemic infection is chronic in mice, and the pathogen persists for many weeks in the organs of untreated animals. Therefore, this model becomes essential for studying humoral and cellular immunological mechanisms during the disease's and host's evolution.

Some studies have shown that caspase–1–dependent cytokines exert essential effects on the initiation of the adaptive Th1 and Th17 cellular responses to fungal infection [32]. Previously, our group showed that IFN-γ is an important cytokine involved in immune defenses during chromoblastomycosis [33–35]; therefore, we speculated that IFN-γ could play a role in the central mechanism regulated by caspase-1 after _F. pedrosoi_ infection. Confirming this hypothesis, the Th1 response, mainly IFN-γ-T-CD4+ cells, was lower in infected _Il-18⁻ᐟ⁻_ mice compared with the wild-type. Therefore, we suggest that the _IL-18⁻ᐟ⁻_ mice were susceptible to chromoblastomycosis because they exhibited reduced Th1 (and not Th17) immunity.

In the context of Th-1 differentiation, caspase-1, via the NLRP3 inflammasome, is required to produce IL-18, which then induces IFN-γ and promotes the efficient killing of the pathogen. Recent studies have shown the importance of IL-18 for the resistance to infection by many pathogens, including _Paracoccidioides brasiliensis_ [36], _Burkholderia pseudomallei_ [37] and _Streptococcus pneumoniae_ [38].

Ketelut-Carneiro et al. [36] showed that _Il18⁻ᐟ⁻_ mice were profoundly vulnerable to paracoccidioidomycosis (PCM), but that, surprisingly, _Il1r1⁻ᐟ⁻_ mice had mortality rates similar to wild-type mice. Thus, the role of IL-18 during PCM is related

to its ability to induce IFN-γ. Our data showed that *Il18*<sup>-/-</sup> mice were more susceptible to infection, which was mediated by the decrease in IFN-γ production. Consistent with our data, IL-18 is upregulated in a caspase–1–dependent manner in response to *C. albicans* and to mediate the development of protective Th1 immunity [32,39].

Understanding the role of IL-18 in the immune response to fungal infections may have implications for developing novel therapeutic approaches, such as targeting IL-18 signaling pathways to modulate the immune response and improve outcomes in patients with fungal infections.

In conclusion, the present data demonstrate, for the first time, that the inflammasome is activated and that NLRP3 and caspase-1 mediated the *F. pedrosoi*-induced production of IL-18, which promotes the Th-1-mediated immune response during chromoblastomycosis.

## Acknowledgments

We thank Maira Cristina Nakamura and Dario Simões Zamboni from the School of Medicine of Ribeirão Preto (University of São Paulo) for providing knockout animals.

## Author contributions

**Conceptualization:** Sandro Rogério Almeida, Lucas Golçalves Ferreira.

**Formal analysis:** Sandro Rogério Almeida, Lucas Golçalves Ferreira.

**Funding acquisition:** Sandro Rogério Almeida.

**Methodology:** Lucas Golçalves Ferreira.

**Supervision:** Sandro Rogério Almeida.

**Writing – original draft:** Sandro Rogério Almeida.

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
