## [Decision Letter · Decision Letter 0]

14 Jan 2025

PONE-D-24-51638IL-18 production is required for the generation of a Th1 response during chromoblastomycosisPLOS ONE

Dear Dr. Almeida,

Thank you for submitting your manuscript to PLOS ONE. After careful consideration, we feel that it has merit but does not fully meet PLOS ONE’s publication criteria as it currently stands. Therefore, we invite you to submit a revised version of the manuscript that addresses the points raised during the review process.

We look forward to receiving your revised manuscript.

Kind regards,

Divakar Sharma

Academic Editor

PLOS ONE

5. Please provide a complete Data Availability Statement in the submission form, ensuring you include all necessary access information or a reason for why you are unable to make your data freely accessible. If your research concerns only data provided within your submission, please write "All data are in the manuscript and/or supporting information files" as your Data Availability Statement.

6. Thank you for stating the following financial disclosure:

“Foundation (FAPESP): Grant FAPESP  2016/04729-3.”

Additional Editor Comments:

Major Revision Requested

Reviewers' comments:

Reviewer's Responses to Questions

**Comments to the Author**

1. Is the manuscript technically sound, and do the data support the conclusions?

Reviewer #1: Partly

Reviewer #2: Partly

Reviewer #3: Yes

2. Has the statistical analysis been performed appropriately and rigorously? 

Reviewer #1: Yes

Reviewer #2: Yes

Reviewer #3: Yes

3. Have the authors made all data underlying the findings in their manuscript fully available?

Reviewer #1: Yes

Reviewer #2: Yes

Reviewer #3: Yes

4. Is the manuscript presented in an intelligible fashion and written in standard English?

Reviewer #1: No

Reviewer #2: Yes

Reviewer #3: Yes

5. Review Comments to the Author

Reviewer #1: Methodology:

Please include a subitem showing the macrophage obtention from BMDM.

correct this statement: intraperitoneal (i.p.) injection of 1 ml of PBS without (the control) or with 2 x 107

It seems that it is unclear to the reader if all experiments were conducted on all types of animals.

Results:

The first result does not mention the mice strain that the macrophages came from as well as the other results.

A better results description showing the mice strain used for macrophages assay is important. The graphs need a better description and indication of the cells used from mice strain and the correct pharmacological approach used indicating with or without inhibitor used.

At the present form important results description is missing.

Reviewer #2: The authors conducted a study with wild mice and mice individually knocked out for NLRP3, IFN-γ, Caspase-1/-11, or IL-18 to verify the role of IL18 and NLRP3 inflammasome in experimental chromoblastomycosis.

The title of the paper should be modified to IL-18 production is required for the generation of a Th1 response during experimental chromoblastomycosis.

The methods are appropriate to achieve the proposed objectives. However, the comparative evaluation of fungal quantification was obtained by counting colony-forming units (CFU) in macerates, liver, and spleen. I suggest that the authors also provide information about the aspects and compare the inflammatory reactions in the liver and spleen of both groups of animals.

Authors must explain what they understand by severe forms of chromoblastomycosis (see page 3, second line of the penultimate paragraph and page 9 - second and third lines of the first paragraph) since in human disease this mycosis usually affects the skin and subcutaneous tissue and rarely manifests as systemic disease. Insert the reference(s) in the first paragraph of page 9.

Insert the reference of the several papers mentioned in the last paragraph of page 9.

Page 15 - Figure 2 caption - Why do the authors classify it as deep chromoblastomycosis? In this model, there is visceral involvement (liver and spleen), therefore systemic disease.

In the conclusion section, the authors should state that their work demonstrates, for the first time, the involvement of immune mechanisms mediated by the NRLP3 inflammasome in experimental chromoblastomycosis.

Reviewer #3: This is an interesting research that demonstrates the role of IL18 in the activation of Th1 response to control chromoblastomycosis infection. Although the expressive presence of such cytokine in human tissue have been previously demonstrated, the direct evidence and interaction in the context of NLRP3 and caspase1 is new.

The last part of page 8, and first topic on page 9 fit better in discussion. In this last one, when considered IFN-gamma, which patients? Is it part of the present research or, maybe it could fit better in discussion?

Page 18, second paragraph, correct NLRP3, not nlrp.

How could authors explain the similar expression of IL10 and IL12?

6. PLOS authors have the option to publish the peer review history of their article (what does this mean? ). If published, this will include your full peer review and any attached files.

**Do you want your identity to be public for this peer review?** For information about this choice, including consent withdrawal, please see our Privacy Policy .

Reviewer #1: No

Reviewer #2: No

Reviewer #3: **Yes: ** Carla Pagliari

---

## [Author Response · Author response to Decision Letter 1]

14 Feb 2025

Please find enclosed the manuscript entitled “IL-18 production is required for the generation of a Th1 response during experimental chromoblastomycosis”, which we re-submit for publication.

We deeply thank the reviewer for the efforts spent in reading our manuscript. As follows, we addressed our replies for the comments. We also highlighted the changes within the manuscript.

Financial Support: This work was supported by São Paulo Research Foundation (FAPESP): Grant FAPESP 2016/04729-3. The funders had no role in study design, data collection and analysis, decision to publish, or preparation of the manuscript.

Reviewer #1:

Methodology:

1-“Please include a subitem showing the macrophage obtention from BMDM.

correct this statement: intraperitoneal (i.p.) injection of 1 ml of PBS without (the control) or with 2 x 107

It seems that it is unclear to the reader if all experiments were conducted on all types of animals.”

Thank you for the suggestion. This statement was improved in the new version of MS.

2-Results:

The first result does not mention the mice strain that the macrophages came from as well as the other results.

A better results description showing the mice strain used for macrophages assay is important. The graphs need a better description and indication of the cells used from mice strain and the correct pharmacological approach used indicating with or without inhibitor used.

At the present form important results description is missing.

Thank you for the observation. We include the mice strain in all assays

Reviewer #2:

1-The title of the paper should be modified to IL-18 production is required for the generation of a Th1 response during experimental chromoblastomycosis.

Thank you for the suggestion. We modified the title as recommend

2-The methods are appropriate to achieve the proposed objectives. However, the comparative evaluation of fungal quantification was obtained by counting colony-forming units (CFU) in macerates, liver, and spleen. I suggest that the authors also provide information about the aspects and compare the inflammatory reactions in the liver and spleen of both groups of animals.

Thank you for the suggestion. We realized the histopathology of organs, but we did not observe a significant difference, so we decided not to include this analysis

3-Authors must explain what they understand by severe forms of chromoblastomycosis (see page 3, second line of the penultimate paragraph and page 9 - second and third lines of the first paragraph) since in human disease this mycosis usually affects the skin and subcutaneous tissue and rarely manifests as a systemic disease. Insert the reference(s) in the first paragraph of page 9.

Thank you for your observation. We clarified in the new version of MS the information about the classification of severity of chromoblastomycosis

4-Insert the reference of the several papers mentioned in the last paragraph of page 9.

We added the references

5-Page 15 - Figure 2 caption - Why do the authors classify it as deep chromoblastomycosis? In this model, there is visceral involvement (liver and spleen), therefore systemic disease.

Thank you for your observation. We change the word deep by systemic

6-In the conclusion section, the authors should state that their work demonstrates, for the first time, the involvement of immune mechanisms mediated by the NRLP3 inflammasome in experimental chromoblastomycosis.

There was already evidence in the literature of the involvement of the NRLP3 inflammasome in chromoblastomycosis. We demonstrated for the first time that this inflammasome was involved in the production of IL-18 and activation of the Th1 response.

Reviewer #3:

1-The last part of page 8, and first topic on page 9 fit better in discussion. In this last one, when considered IFN-gamma, which patients? Is it part of the present research or, maybe it could fit better in discussion?

Thank you for the suggestion. The paragraph commented on was moved to the discussion. Concerning IFN-g, we talk about a mild form of chromoblastomycosis

2-Page 18, second paragraph, correct NLRP3, not nlrp.

The word was corrected

3-How could authors explain the similar expression of IL10 and IL12?

At certain moments in the evolution of the disease, as in chromoblastomycosis, there is no cytokine predominance; thus, it is possible to observe both cytokines being secreted in a mixed system of cytokine-producing cells.

---

## [Decision Letter · Decision Letter 1]

18 Mar 2025

IL-18 production is required for the generation of a Th1 response during experimental chromoblastomycosis

PONE-D-24-51638R1

Dear Dr. Almeida,

We’re pleased to inform you that your manuscript has been judged scientifically suitable for publication and will be formally accepted for publication once it meets all outstanding technical requirements.

Kind regards,

Divakar Sharma, Ph.D.

Academic Editor

PLOS ONE

Additional Editor Comments (optional):

Accept

Reviewers' comments:

Reviewer's Responses to Questions

**Comments to the Author**

1. If the authors have adequately addressed your comments raised in a previous round of review and you feel that this manuscript is now acceptable for publication, you may indicate that here to bypass the “Comments to the Author” section, enter your conflict of interest statement in the “Confidential to Editor” section, and submit your "Accept" recommendation.

Reviewer #1: All comments have been addressed

Reviewer #2: All comments have been addressed

2. Is the manuscript technically sound, and do the data support the conclusions?

Reviewer #1: Yes

Reviewer #2: Yes

3. Has the statistical analysis been performed appropriately and rigorously? 

Reviewer #1: Yes

Reviewer #2: Yes

4. Have the authors made all data underlying the findings in their manuscript fully available?

Reviewer #1: Yes

Reviewer #2: Yes

5. Is the manuscript presented in an intelligible fashion and written in standard English?

Reviewer #1: Yes

Reviewer #2: Yes

6. Review Comments to the Author

Reviewer #1: The authors provided the answers adequately.

I think the manuscript can be accepted for publication.

Thank you for this opportunity.

Reviewer #2: The authors responded and answered the questions appropriately. Nothing more to comment.

7. PLOS authors have the option to publish the peer review history of their article (what does this mean? ). If published, this will include your full peer review and any attached files.

**Do you want your identity to be public for this peer review?** For information about this choice, including consent withdrawal, please see our Privacy Policy .

Reviewer #1: No

Reviewer #2: **Yes: ** Mirian Nacagami Sotto

---

## [Editor Report · Acceptance letter]

PONE-D-24-51638R1

PLOS ONE

Dear Dr. Almeida,

I'm pleased to inform you that your manuscript has been deemed suitable for publication in PLOS ONE. Congratulations! Your manuscript is now being handed over to our production team.

Kind regards,

on behalf of

Dr. Divakar Sharma

Academic Editor

PLOS ONE